# Patient-Derived Lung Tumoroids—An Emerging Technology in Drug Development and Precision Medicine

**DOI:** 10.3390/biomedicines10071677

**Published:** 2022-07-12

**Authors:** Hélène Lê, Joseph Seitlinger, Véronique Lindner, Anne Olland, Pierre-Emmanuel Falcoz, Nadia Benkirane-Jessel, Eric Quéméneur

**Affiliations:** 1INSERM (French National Institute of Health and Medical Research), UMR 1260, Regenerative Nanomedicine, CRBS, 1 Rue Eugène Boeckel, 67000 Strasbourg, France; hle@transgene.fr (H.L.); jo.seitlinger@gmail.com (J.S.); veronique.lindner@chru-strasbourg.fr (V.L.); anne.olland@chru-strasbourg.fr (A.O.); pefalcoz@gmail.com (P.-E.F.); nadia.jessel@inserm.fr (N.B.-J.); 2Transgène SA, 400 Boulevard Gonthier d’Andernach, 67400 Illkirch-Graffenstaden, France; 3Faculty of Medicine and Pharmacy, University Hospital Strasbourg, 1 Place de l’Hôpital, 67000 Strasbourg, France

**Keywords:** preclinical models, non-small-cell lung cancer, tumoroids, microfluidic

## Abstract

Synthetic 3D multicellular systems derived from patient tumors, or tumoroids, have been developed to complete the cancer research arsenal and overcome the limits of current preclinical models. They aim to represent the molecular and structural heterogeneity of the tumor micro-environment, and its complex network of interactions, with greater accuracy. They are more predictive of clinical outcomes, of adverse events, and of resistance mechanisms. Thus, they increase the success rate of drug development, and help clinicians in their decision-making process. Lung cancer remains amongst the deadliest of diseases, and still requires intensive research. In this review, we analyze the merits and drawbacks of the current preclinical models used in lung cancer research, and the position of tumoroids. The introduction of immune cells and healthy regulatory cells in autologous tumoroid models has enabled their application to most recent therapeutic concepts. The possibility of deriving tumoroids from primary tumors within reasonable time has opened a direct approach to patient-specific features, supporting their future role in precision medicine.

## 1. Introduction

Lung cancer is the deadliest cancer worldwide; non-small cell lung cancer (NSCLC) is the most common form, with 85% of all cases [1]. The survival rate over 5 years for patients with advanced stage lung cancer remains below 15% despite the diversity of therapeutic treatments and very important progress in the last two decades [2]. Treatment options mainly rely on surgery, complemented with radiotherapy, targeted chemotherapy, or immunotherapy, thanks to the development of specific markers of response [3]. The 5 year survival rate improves to 61.2% when diagnosis is performed at the stage of localized tumor, but drops to 9.9% when cancer is detected at the metastatic stage [4]. Both the large National Lung Screening Trial (NLST) conducted in the U.S. from 2002 to 2011, and the NELSON study in Europe, confirmed that the earlier the diagnosis, the higher the survival rate [5,6]. However, about 70% of lung cancer patients remain diagnosed at advanced stages, where heavy systemic treatment is necessary [7]. 

For patients at advanced stages, platinum-based chemotherapy regimen is the standard of care but is associated with severe toxicities [8]. In this respect, therapies targeting driver mutations in EGFR, or more recently, KRAS genes, have been a progress, but often face occurrence of resistance [9,10]. Immunotherapy has recently revolutionized the treatment of lung cancer [11]. The main strategy in immunotherapy is to target immune checkpoint pathways in order to escape local immune tolerance and to boost anti-tumor response [12]. Its therapeutic window is quite narrow, and the use of immune-checkpoint-blockers is associated with a high rate of immune-related adverse events (irAEs), reaching 26.82% of patients treated with PD-1 inhibitors [13]. These deleterious responses might affect multiple organs (skin, digestive tract, liver, endocrine gland, lung, thyroid, etc.) [14,15,16]. 

To better predict and assess the efficacy, resistance or toxicity of drug candidates during drug development, experimental advanced models have been designed. The most relevant are 3D models derived from both normal and tumoral lung epithelial cells. They aim at recapitulating the heterogeneity of tumoral cells, and at reproducing the complex network of interactions in the tumor microenvironment (TME). Taking advantage of the fast evolution of cell culture technologies and microsystems, 3D tumor models represent a major step forward in the characterization of new drug candidates and pave the way towards personalized medicine by using patient-derived tumor biopsies. This review aims to describe state-of-the-art lung cancer preclinical models, their limits in predicting drug efficacy in complex tumors or adverse events, and recent technological progresses that might rapidly benefit the search for safe and efficient drugs (Figure 1). 

## 2. Current Experimental Models for Drug Development in NSCLC and Their Limits

### 2.1. Lung Cancer Cell Lines for In Vitro Studies

Lung cancer cell lines have been widely used in cancer research, the “historical” A549 cell counting itself 17,947 entries in PubMed (as of 17 March 2022). The NCI panel of cancer cell lines comprises more than 200 lung cancer cell lines derived from patients with either small-cell lung cancer (SCLC) or NSCLC [17,18]. Whereas monotypic cell cultures are suitable for the study of proliferative mechanisms and the study of signaling pathways, they have proven insufficient to understand some major interactions within the TME (e.g., with stromal cells, endothelial cells and/or immune cells). Pro-inflammatory cells and stromal cells were shown to be key in controlling tumor growth, metastasis and angiogenesis [18]. Another issue with the use of cancer cell lines is genetic variation; many authors have documented the loss of original phenotypic features from the primary tumor [19,20]. Despite these drawbacks, lung cancer cell lines still feed the vast majority of basic studies in cancer research, and of early drug screening campaigns [21].

### 2.2. Murine Models for In Vivo Studies

(a)Patient-derived tumor xenografts (PDXs)

PDXs have been used for understanding cancer metastasis and for drug screening. Biopsies and patient-derived tumor materials offer the advantage of encompassing multiple factors such as cellular heterogeneity, histological structures, malignant genotypes and phenotypes. Grafted onto immunodeficient mice, they tend to conserve essential features of the human primary tumor. In particular, somatic and genomic alterations and histological subtypes were found to be comparable between primary tumors and corresponding PDX [22,23,24]. Nevertheless, major limitations are reported. Genomic variation seems higher in PDXs, with an enrichment of aberrations in cancer associated genes [22]. Immunodeficient NOD/SCID or NOD/NSG mice are still largely used to avoid tumor rejection but they are not suitable for the assessment of immunotherapies [23]. Humanized PDX models are thus recommended in this perspective but are very expansive. The question of implantation site is also important, orthotopic grating or injection into the circulation are associated with a higher success rate, up to 30–40%, than subcutaneous grafting [25].

(b) Syngeneic murine models

These immunocompetent models turned out to be essential for understanding both tumor-host interactions and immune mechanisms [26]. Unfortunately, there is still a limited panel of murine lung cancer cell lines that can spontaneously form tumors in immunocompetent mice [27]. The development and validation of relevant immunocompetent syngeneic models for lung cancer will be a long process. 

(c) Genetically engineered mouse models (GEMMs)

GEMMS were designed to approach genetic characteristics of human tumors that cannot be reflected in xenograft models, allowing disease modeling in immunocompetent environments. They are inducible models, enabling either overexpression, shutoff or functional replacement of selected genes of interest [17]. Lung cancer GEMMs targeting oncogenic drivers, such as KRAS or EGFR, are available for assessing response to targeted therapies, and discovering new pathways implicated in malignancy [28]. Resistant models to EGFR inhibitors were also reported [29]. A major hurdle in their development is that the establishment of GEMMs is rather expensive and long. Furthermore, validation of experimental procedures is important, as evolution of GEMMs might be highly variable within a cohort. Last, but not least, tumors with low malignancy may fail to recapitulate the tumor–host interactions in the course of cancer progression. 

## 3. Limits of Current Preclinical Models in Lung Cancer Research

Current preclinical models fail to effectively mimic human responses [30]. These limitations, that impact both basic understanding of human tumor biology, as well as drug development processes, are summarized in Table 1. Major problems are: (i) murine stromal components replacing their human counterparts, (ii) the lack of immune system in most models, and (iii) the lack of the many interactions that characterize a fully functional TME [31]. Interestingly, the recent progress in 3D cultures of human cancer cells might help to overcome these limitations. 

Interestingly, the recent progress in 3D cultures of human cancer cells might help to overcome these limitations. Numerous advantages have been reported compared with regular preclinical models, recapitulating complex structural features of natural tumors, and making them more predictive of patients’ individual responses. They also retain cancer general features such as hypoxia or necrotic domains, or substructures of drug resistant cells [32]. The 3D tumor models directly benefit from the large R&D effort in developing organoids and next-generation preclinical models, matching the ethical standards associated with the 3R approach. 

## 4. Tumoroids: A Next-Generation Preclinical Model

Table 2 highlights representative examples of recent progress in the field. Kim et al. demonstrated that lung tumoroids can retain specific histological features of the primary tumor, as well as spontaneous TP53 and EGFR mutations [24]. The closely related concept “tumor-like organoids”, in other words, tumoroids, has spread to numerous laboratories [24,33,34,35,36,37]. All the 3D models cited use primary cells as starting material, with the aim of representing heterogeneity inter-patients, to better understand patient-specific drug responses.

Among these different 3D models of lung cancer, we can observe a discrepancy in the definitions, that may lead to misunderstanding between the terms of “spheroids”, “organoids” and “tumoroids”. Spheroids are a monotypic cell system that concentrate more in structure than functionality. They retain less of the tissue architecture, compared with organoids, that represents the functionality of a healthy organ [46,47]. Similar to organoids, which are 3D self-organized cultures of organ-derived cells recapitulating major physiological functions, tumoroids are functional surrogates of native tumors. Derived from patients’ tumoral tissues, they have become widely used in understanding molecular pathways of carcinogenesis, in drug development and personalized medicine [35,47]. The nomenclature of the 3D models should be harmonized, and to make this clearer, we will refer to “Patient-derived tumoroids” for tumoral cells derived from patients that structurally and functionally represent the pathology.

Nevertheless, some limitations can be noted, including the lack of stromal and immune cells in the TME. The development of a relevant model for immuno-oncology is still needed [48].

## 5. Towards the Optimization of Lung Tumoroids

Several teams, including ours, have developed cancer models based on patient-derived tumoroid, and incorporating relevant healthy cells (fibroblasts, macrophages) known to contribute to the overall function of the tumor. Figure 2 provides an example of such a model where lung tumoroids are combined with CAFs. The patient-derived material is cultured within 1 week after biopsy or surgery, in order to fit with the delay of 6 weeks usually associated with the therapeutic decision. 

In a representative example, the lung tumoroid very well reproduced the histological features of the primary tumor, an acinous adenocarcinoma (Figure 3). Anatomopathological analysis was positive staining for thyroid transcription factor 1 (TTF-1) and negative for p40, two markers regularly used to distinguish adenocarcinoma from squamous cell carcinoma. TTF-1 and Ki-67 staining confirmed the proliferative character of the tumoral cells. Mucin-1 (MUC-1) and cytokeratin-7 (CK-7) were also present in lung epithelia and alveoli. CK-7 was expressed in 94–100% of lung adenocarcinoma [49]. After 1 week of culture, we observed enrichment in immune cells. This opens the door to immunophenotypic analysis, and immunopharmacological studies.

## 6. Application to Drug Discovery, Screening, and Study of Mode of Action

The use of tumoroids is expanding, and their utility for basic research and early steps of drug development has been recognized. They have demonstrated their ability to recapitulate the histology and genetic characteristics of the primary tumor [24]. Li et al. established a biobank of PDOs of EGFR-mutated tumoroids for the assessment of targeted therapies. In another example, Zhang et al. reported that cisplatin was less effective in PDOs generated from NSCLC tissues than from cell lines, thus demonstrating that patient-derived material should be preferred and can inform on important resistance mechanisms [38]. Interpatient heterogeneity is another important parameter to take into account during drug development. In a study comparing two patients that harbored the same EGFR p.L858R mutations, and naïve to any prior treatment, Kim et al. demonstrated that erlotinib, a tyrosine kinase inhibitor, acted differently. The resistant patient displayed amplification of c-Met protein and was thus eligible to crizotinib [24]. This example shows that mutational profiling benefits from complementary experimental procedure. Predictive drug assessment involves the use of an in vitro model with well-known drugs in order to model clinical responses, and they are now available for large screening campaigns. Li et al. screened a large panel of 12 anticancer drugs in their model of lung adenocarcinoma organoids, and showed a large heterogeneity of measurable responses [35]. Tumoroids will also constitute a good model for testing new treatment perspectives such as photothermal therapy based on near-infrared irradiation [50,51]. Thus, the use of patient-derived tumoroids has demonstrated success in predicting patients’ responses and constitute physiologically relevant oncology models for drug testing and drug discovery [52].

Omics technologies are now standard approaches to study patient molecular heterogeneity for the purpose of precision medicine [53]. Tumoroid technologies successfully combine with omic analysis. For instance, Xu et al. used proteomic to decipher drug resistance mechanisms in a human metastatic lung cancer-derived cell line cultured on a multi-organ microfluidic chip. The cell viability assay confirmed that metastatic cells presented acquired resistance to chemotherapy and targeted therapies, and omic analysis established this was due to an increase in DNA replication and glutathione metabolism [54]. Li et al. developed a patient-derived tumoroid assay for drug screening and used whole exome sequencing to explain the responses to therapies; in particular to afatinib for EGFR-mutated patients [35,55]. Peng et al. used transcriptomics to compare primary tissues of squamous carcinoma or adenocarcinoma to their corresponding tumoroids. Tumoroid models displayed a higher transcriptional accuracy compared with PDXs or cancer cell lines. This reliability was attributed to the 3D structures and spontaneous self-organization [56]. Ma et al. explored the potential of LUAD and LUSC organoids with the use of transcriptomic analysis for the identification of genes implicated in NSCLC tumorigenesis process. Transcriptomic analyses were performed both on tumoral tissue and normal tissue, and also on matching tumoral and healthy organoids. They constitute an interesting study where they discovered the expression of three genes (CDK1, CCNB2 and CDC25A) that may predict a poor prognosis in an adenocarcinoma patient [57]. Furthermore, their finding on the CDK1 was supported by Wang et al. who found its implication in lung cancer progression and development. Similarly, Wang et al. studied the CCNB2 gene by knocking it down and observed an inhibition of tumoral growth in some adenocarcinoma cell lines (2D cultures) [58].

## 7. Modelling the Tumor Microenvironment (TME)

TME is implicated in tumorigenesis and tumor progression, and has a strong influence on drug response, either positively or negatively. It might explain some difference of drug sensitivity usually observed between models [59]. Overall, TME controls immunocompetence, as nicely illustrated by Finnberg et al., in their study comparing tumor-infiltered immune cells and blood immune cells. An increased proportion of myeloid-derived suppressor cells (mDSCs) and a decreased proportion of NK cells and monocytes in a tumor site are hallmarks of an immunosuppressive microenvironment that would not respond to immunotherapy [60]. A tumor’s interactions with the immune cells as illustrated in Figure 4 shows their influence toward lung cancer progression [61].

The introduction of immune-checkpoint inhibitors (ICIs) targeting programmed death protein (PD-1) and its ligand (PD-L1) or cytotoxic lymphocyte antigen 4 (CTLA-4) has improved the overall survival of advanced NSCLC patients. However, it should be kept in mind that only 20% of patients are responders to ICIs in advanced NSCLC, and mechanisms of resistance are not yet fully elucidated [62,63]. Several ICIs have been approved since 2015 for advanced NSCLC patients, but still, adverse effects or non-responder profiles are observed (Table 3). To understand patients’ heterogeneity that are observed at the clinical level, the format of tumoroids seems particularly relevant for modelling the immune cells of the TME. This kind of model will help to understand how tumor and immune cells contribute to therapies’ responses [64,65]. 

Thus, an autologous immune cell population in the tumoroid model will facilitate the identification of patients who are likely to develop primary or acquired resistance. Primary resistance is characterized by two parameters: (i) tumor-infiltrating lymphocytes (TILs) and (ii) PD-L1 expression as defined in the Immunoscore. Galon et al. have pioneered the assessment of CD3+ and CD8+ TILs in colorectal cancer. Following their studies, they created the Immunoscore, which is defined by the density of CD3+ and CD8+ T cells (low, intermediate or high) in tumor stroma, invasive margin and tumor center [73]. The prognostic value of TILs was demonstrated in patients with colorectal cancer, but not yet for lung cancer.

Dijkstra et al. developed a co-culture of autologous T cells with NSCLC organoids, where they convincingly demonstrated specific tumor reactivity with the use of peripheral cells, showing the possibility of recreating an important function of the TME, namely, antigen presentation [74]. Very few studies have dealt with immune-competent organoids either in NSCLC or any other therapeutic areas. In a co-culture of PDOs and THP-1, human monocytes (that differentiate into macrophages after stimulation with IFNγ) were developed [75]. In this model, trastuzumab induced a THP1-dependent cytolysis on PDOs that was identified as an antibody-dependent cell-mediated cytotoxicity (ADCC). In the same study, nivolumab and pembrolizumab were assessed in the presence of enteroxin-stimulated PBMCs. Both ICIs induced a higher percent cytolysis. In another study, melanoma patient-derived explants were co-cultured in the presence of CD8+ T cells, Tregs and macrophages, to reproduce T-cell mediated tumor lysis and its regulation [76]. In order to test ICIs in vitro, some studies analyzed whether PD-L1 expression in tumoroids correlated with that measured in primary tumors [77]. 

An original approach to generate an immune-competent tumor model in vitro was to connect a patient-derived tumoroid with an artificial lymph node containing mature antigen-presenting cells. The tumoroid microsystem was shown to maintain functional class I and class II cross-presentations of antigens enhanced with organoids produced from a patient’s lymph nodes and was shown to maintain functional class I and class II cross-presentations of antigens [78]. This was for melanoma; we are not aware of such development in lung cancer research.

## 8. Looking for Biomarkers in Lung Cancer Therapeutic Management and Lung Cancer Progression

The identification of molecular alterations in NSCLC has opened new therapeutic options. About 60% of lung adenocarcinoma and 50–80% of squamous cell carcinoma present a known oncogenic driver mutation [9]. Different types of mutations have been identified, five amongst them being considered as clinically relevant biomarkers: EGFR, ALK, MET, ROS-1 and KRAS. For NSCLC patients at advanced stages, the monitoring of PD-L1 expression is also recommended by clinical guidelines to orient the treatment with either anti-PD-1 or anti-PD-L1 [79]. The field of therapeutic biomarkers has greatly improved but research is still required; most NSCLC patients do not present an actionable mutation and high PD-L1 expression is only detected in 29.5% of them [80]. However, the predictive value of PD-L1 % was brought into question as many patients with low expression of PD-L1 were shown to respond to nivolumab in the Checkmate 063 clinical study. The FDA approved pembrolizumab in first-line treatment for metastatic non-squamous NSCLC in combination with pemetrexed and carboplatin regardless of PD-L1 expression [81,82].

Tumoroid technologies constitute interesting and affordable tools for biomarker discovery in the era of precision medicine [34]. We previously referenced the work by Li et al. who showed how patient-derived tumoroids allowed the identification of four genes (RHOF, SLC16A3, ANXA10, CDHR1) as predictive biomarkers for survival [35]. 

## 9. Convergence of Technologies into Microfluidic Systems

Tumoroids have shown their maturity in conventional applications, but combination with microfluidics allows dynamic control of the TME that further expands their potential [83]. By providing a direct control of physical conditions (temperature, pH, nutrient and oxygen supply and waste elimination), microfluidics greatly improves the culture of tumoroids. Moreover, this controlled environment allows researchers to more closely attain in vivo conditions, mimicking the venous and lymphatic draining. A combination of sensors in the chip format enables more precise monitoring and environmental lymphatic draining infusion and control. Hypoxia is an essential regulator of the TME function, promoting tumor angiogenesis. Hypoxia involves activation of HIF-1α that induces secretion of VEGF-A. Abnormal vasculature is a consequential damage that affects antitumor efficacy. Indeed, homing of immunocompetent cells is hindered by the development of abnormal vasculature. Thus, microfluidic systems could largely improve our understanding of immune cells in the TME, and efficacy of immunotherapeutic candidates, compared with static models based on co-culture of tumoroids and immune cells [84,85]. With microfluidic chips, it is also possible to study the characteristics of tumor vasculature including tumor angiogenesis and metastasis that are part of tumor physiopathology [86].

Responses to drugs largely differ between static conditions and fluidic conditions. Yildiz-Ozturk et al. studied the effect of panaxatriol on 3D spheroids in static conditions versus fluidic conditions (applied flow rate was 2 µL/min) and they observed that panaxatriol was more effective in fluidic systems. Microfluidic better mimics in vivo conditions and will increase the predictivity of new potential drugs for NSCLC patients [87]. Microfluidics offer a dynamic drug screening as described by Schuster et al.; real-time monitoring of the drugs effect is possible with imagery, drugs’ exposure time can be controlled, continuous flow or fixation of drugs’ pulses and combination drugs’ regimen can be applied (including sequential regimen) [88]. 

Advanced tumoroid models can help immunotoxicology. As previously mentioned, immune-related adverse events (irAEs) can occur in most patients under ICI treatment. These irAEs can affect multiple organs systems (skin, digestive tract, liver, endocrine gland, lung, thyroid, etc.) that are now well modeled in organoid technology. With the implementation of these different model organs on-a-chip, their link with intratumoral mechanisms can be approached. The development of multi-organs-on-chips to study ICI toxicity represents a great solution, helping clinicians in therapeutic management [84]. These multi-organs-on-chips are obviously important in drug discovery, and provide supplementary data for the development of safer and more efficient drugs [89].

Hassell et al. designed a chip by gathering elements from the TME, such as epithelial cells, endothelial cells and two compartments reproducing mechanical breathing motions. They could successfully analyze the functions of the T790M mutation responsible for resistance to the third generation of tyrosine kinase inhibitors, which was not possible under static culture conditions. The system also reproduces the reduction in IL-8 levels upon TKI treatment, that was observed during clinical development, but never observed under static culture conditions [90]. Ruppen et al. cultured primary lung tumor cells alone or in co-culture with pericytes on-a-chip and found that pericytes formed a protective barrier against cisplatin treatment [91]. These recent works support the value of microfluidics, and tumoroid-on-chip technology to model resistance mechanism. From the perspective of precision medicine, it offers the advantage of making maximum use of a patient’s sample by combining different assays such as histology, imagery, cell-based assays, genomic, transcriptomic and proteomic. Miniaturized systems allow smaller samples and multiplex analysis, among others. The field of biosensing opens up detections of a large number of biomarkers. A microfluidic chip integrated with a biosensor for the detection of cytokeratin fragments from NSCLC in the order of 0.1 pg/mL to 100 ng/mL was developed by Feng et al. to study cancer biomarkers for disease diagnosis and prognosis [92,93]. 

Microfluidics can help to decipher many interrogations concerning the lung cancer physiopathology or the drugs’ assessment; these applications are summarized in Table 4.

Figure 5 summarizes this fast-evolving environment that will directly improve the representativity of patient-derived tumoroids, and favor their use in pharmacology, toxicology, and precision medicine.

## 10. Discussion and Perspectives

We predict therapeutic innovation and precision medicine will be the main drivers of the tumoroid technology. The attrition rate for anti-cancer drugs has been two to four times higher than for other drugs in the period from 1979 to 2014 [94]. For NSCLC indication, in particular, only 10 anticancer drugs were approved by the U.S. FDA compared with 25 in breast cancer indication, due to the clinical success of hormonotherapy. Therefore, to decrease this high attrition rate, being 92% for NSCLC drugs, the solution largely relies on the progress and refinement of preclinical models, including tumoroids [95]. 

However, the field is still in its emergence phase, with many researchers developing their own tumoroid prototypic models. This results in large variability among laboratories, and for the various factors to consider: tumor sampling, culture conditions, co-culture conditions, bioanalytical methods, quality control, etc. 

As tumoroids become more widely recognized, standardization and harmonization of the protocols for a validated procedure will help regulatory acceptance. Once standardized, tumoroids and microfluidics models may undoubtedly find their place in all steps of drug development, from target discovery, drug testing, polymorphism analysis, to predictive toxicology [96]. 

We suggest that tumoroids and microfluidics models have a key role to play in the transition from preclinical to clinical phases. Sura et al. highlighted the role of pathologists in these next-generation models, as they already play an important role in animal models [97]. Bioengineers, cell biologists, immunologists, physicists, and physicians are also expected to work together to overcome the current challenges. In this first review, which has focused on lung cancer and highlighting some pioneering studies, we aimed to demonstrate the fast emergence of patient-derived tumoroids, and its natural integration within the most innovative research technologies. The field has not reached its maturity, with the best certainly yet to come. 

## Figures and Tables

**Figure 1 biomedicines-10-01677-f001:**
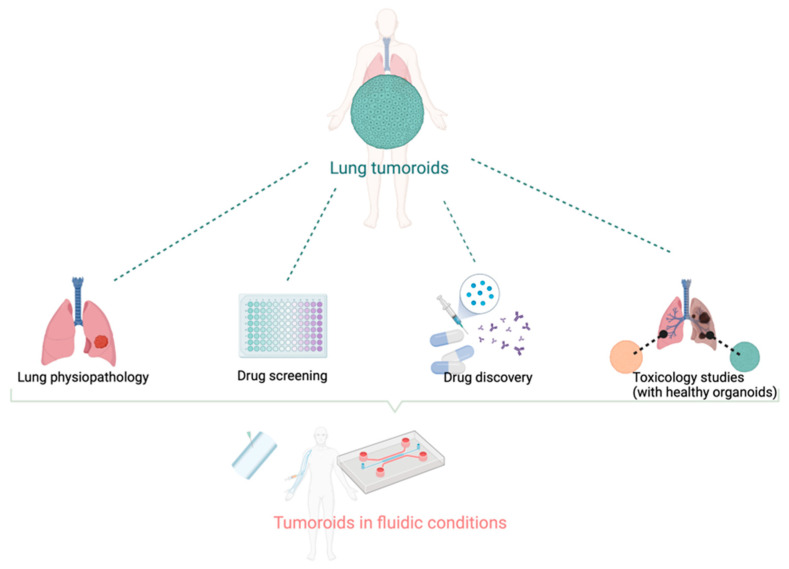
**The variety of applications for tumoroids and organoids models**. Directly derived from patient biopsy or surgery, synthetic tumor models can be used to study tumorigenesis, tumor growth, and interactions with the normal tissues. The drug discovery process should benefit from higher predictivity from these models than current preclinical models. The combination with microfluidic systems allows for better mimicry of the tumor dynamics and makes these tumor models suitable with high throughput/high content bioassays. Created with BioRender.com.

**Figure 2 biomedicines-10-01677-f002:**
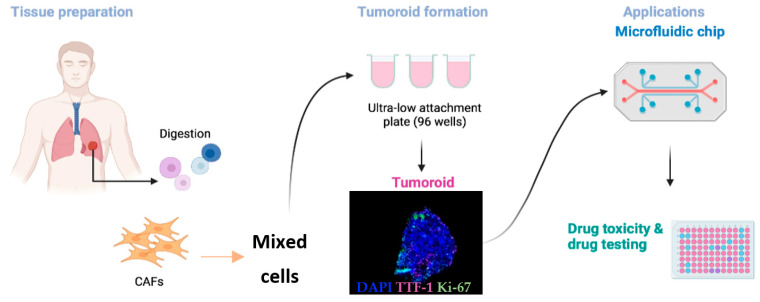
Example of NSCLC tumoroid model and its applications. Tumor samples were obtained from NHC (Nouvel Hôpital Civil) in Strasbourg, France. Washing and enzymatic digestion were performed within 1–2 h after resection. Tumoral cells were centrifuged, and other cell types, such as cancer-associated fibroblasts (CAFs) were added to reconstitute the TME complexity. Tumoroids were grown in a final volume of 200 μL in ultra-low attachment 96-well plates, forming after 5 days of incubation at 37 °C. This tumoroid model is aimed at being inserted into a microfluidic chip to overcome the limit of the static 3D organoid model. Drug toxicity and efficacy assessment are facilitated to select the best treatment opportunity for each individual patient. Created with BioRender.com.

**Figure 3 biomedicines-10-01677-f003:**
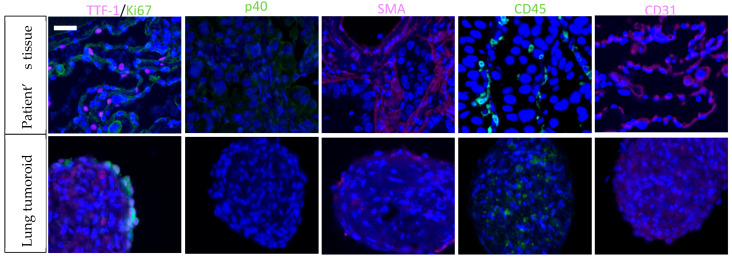
Immunohistochemical features of a patient-derived tumoroid and matched tissue. Patient 2OT 360 suffered from an advanced adenocarcinoma. Scale bar 100 µm.

**Figure 4 biomedicines-10-01677-f004:**
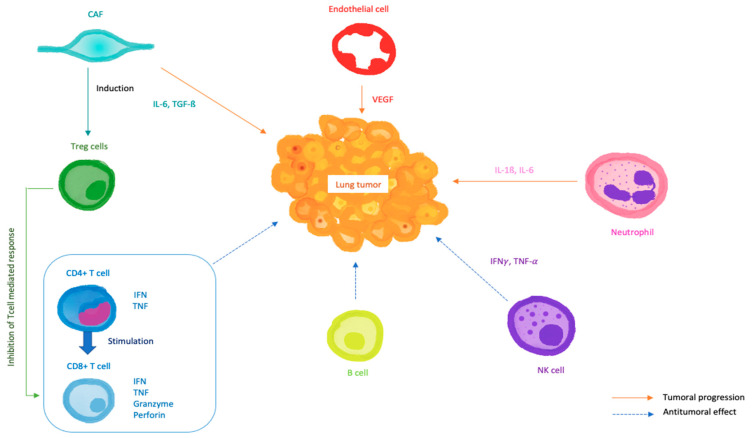
Functional interactions between tumor cells and the main cells in the lung tumor micro-environment. Lung tumor micro-environment is defined by complex interactions between several actors including CAFs, endothelial cells and immune cells. CAFs promote an immunosuppressive microenvironment by interacting with Treg cells, and tumorigenesis by releasing inflammatory cytokines such as IL-6 and TGF-β. Tumorigenesis is also promoted by neutrophils with the release of inflammatory cytokines including IL-1β. On the other hand, CD4+ and CD8+ T cells, B cells and NK cells coordinate an antitumor response. Drawing created using the Tayasui Sketches application.

**Figure 5 biomedicines-10-01677-f005:**
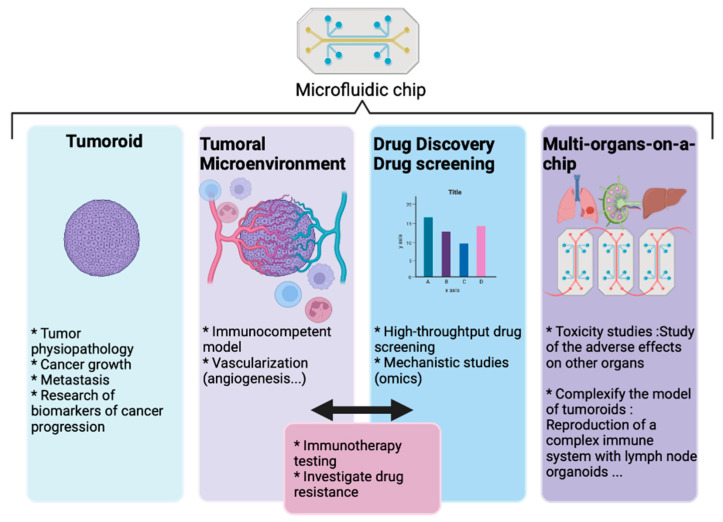
Integrative microfluidics. A combination of tumoroid and microfluidic technologies into a chip format provides access to more relevant, and multipurpose models. Created with BioRender.com.

**Table 1 biomedicines-10-01677-t001:** Advantages and limits of the main preclinical lung cancer models.

Technologies		Advantages	Limits	References
** *In vitro* **	Cancer cell lines	Pure population of tumor cells Replicative ability Large diversity of genomic backgrounds	- Clonal cells poorly reflect the patients’ primary tissue - Genomic instability - Absence of stromal, endothelial, and immune cells	[17,18,19,20]
** *In vivo* **	PDXs	Closer to patients’ primary tissues	- Genetic heterogeneity and epigenomic instability - Lack of TME components - Immunodeficiency - Variable implantation, instability	[17,22,23,24,25]
	Syngeneic models	Functional immune system	- Poor clinical prediction - Tedious process	[27]
	GEMMs	Functional immune system Inducible model	- Long process - Low malignancy potential because of the long latency period	[17,28,29]

**Table 2 biomedicines-10-01677-t002:** Current lung organoid/tumoroid models.

Primary Tumor Histology, (Mutations) *	Technology Name	Culture Time	Applications	Ref.
NS	Patient-derived tumor spheroid (PDS)	120 days	Mechanistic studies Resistant models Drug screening	[38]
ADK, SCC, LCC	Lung cancer organoids	>1 year	Drug screening	[39]
ADK, SCC, LCC	Patient-derived lung cancer organoids	>6 months	Patient-specific drugs screening Living biobank as support to xenograft model	[24]
ADK, SCC	NSCLC organoids	3 months	Drug screening	[37]
ADK, SCC NSCLC (EGFR, KRAS)	Patient-derived organoids models (PDOs)	NS	Genomic analyses Production of treatment response	[40]
NSCLC (EGFR, KRAS)	Patient lung-derived tumoroids (PLDTs)	NS	Drug screening	[41]
ADK, SCC, LCC, NSCLC	Lung cancer organoids	NS	Personalized medicine	[36]
NS	Patient-derived organoids (PDOs)	2–3 months	Drug screening Comparative analysis	[42]
ADK	Lung ADK (LADC)-derived organoid model	>50–200 days	Transcriptome analysis Biomarkers discovery Drug screening Living biobank	[35]
ADK and SCC	Lung cancer organoids	6 days	Drug screening	[43]
ADK and SCC primary or metastatic NSCLC	Patient-derived tumoroids (PDTs)	>13 months	Generation of cell lines	[44]
ADK	Patient-derived tumoroids (PDTs)	4 days	Mimic the tumor vascular network PDTs ready to use in microfluidic device for drug screening	[45]

*** ADK:** adenocarcinoma; **SCC:** squamous cell carcinoma; **LCC:** large cell carcinoma, **NS**: non specified.

**Table 3 biomedicines-10-01677-t003:** ICIs approved and on ongoing clinical evaluation for advanced NSCLC patients.

Drug	Target(S)	Indications	FDA Approval	Ref.
Pembrolizumab KEYTRUDA^®^	PD-1	First-line systemic therapy for NSCLC patients with PD-L1 expression > 50% and without EGFR or ALK mutations Second-line advanced stage NSCLC after progression on first-line chemotherapy (PD-L1 > 1%) First-line for metastatic non-squamous NSCLC in combination with pemetrexed and carboplatin (regardless of PD-L1 expression)	2015 2015 2017	[66] [67]
Nivolumab OPDIVO^®^	PD-1	Advanced squamous and non-squamous NSCLC as second-line systemic therapy after progression on first-line chemotherapy (regardless of PD-L1 expression) First-line treatment for metastatic or recurrent NSCLC without EGFR or ALK mutations, in combination with Ipilimumab YERVOY^®^ (anti CTLA-4) and 2 cycles of platinum-doublet chemotherapy	2015 2020	[68] [69]
Durvalumab IMFINZI^®^	PD-L1	Unresectable stage III NSCLC patients that have not progressed after chemoradiation therapy	2018	[70]
Atezolizumab TECENTRIQv	PD-L1	First-line treatment in metastatic NSCLC with PD-L1 > 50%	2020	[71]
Cemiplimab-rwlcLIBTAYO^®^	PD-1	First-line treatment in locally or metastatic advanced stage NSCLC (no eligible to surgical resection nor definitive chemoradiation) with PD-L1 > 50%	2021	[72]

**Table 4 biomedicines-10-01677-t004:** Summary of integrative microfluidics systems in NSCLC.

Microfluidic Model	Applications	Ref.
Microfluidic device for lung cancer organoids	Drug screening ⇨ Predictive modelling of chemotherapy response (cisplatin and etoposide) ⇨ Presence of chemo-resistant cells in the inner core of organoids	[83]
Lung carcinoma spheroid based microfluidic platform	Drug assessment of panaxatriol in fluidic conditions with a perfusion function on cancer cells and healthy cells	[87]
Human organ chip model	Recapitulation of human cancer with its specific microenvironment Assessment of tyrosine kinase inhibitors’ (TKI) responses to physical cues mimicking breathing motions ⇨ TKI resistance is modelled on a lung-cancer-on-a-chip with breathing motions	[90]
Lung cancer cell spheroids in a perfused microfluidic platform	Cell viability assessment of chemotherapeutical drug	[91]
Detection of cytokeratin 19 fragments	Biomarkers study of diagnosis and prognosis	[93]
Chip for study of lung cancer brain metastasis	Study metastasis	[54]

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
