# Peer review of "Patient-Derived Lung Tumoroids—An Emerging Technology in Drug Development and Precision Medicine"

_biomedicines, 2022, doi:10.3390/biomedicines10071677_

Round 1

Reviewer 1 Report

The review “Patient-derived lung tumoroids an emerging technology in drug development and precision medicine” explores the state of the art and the applications of lung tumoroids in clinical research. The organization of the review is very schematic, allowing a comprehensive understanding of the subject. I do recommend this paper for publication after minor revision.

-          Resolution of figures 1 and 4 could be improved;

-          A table including the state of the art in integrative microfluidics could help focusing on current research;

-          The section in which immunotherapy is discussed (7) should be expanded;

-          The use of photothermal therapy, which is a non-invasive technique capable  of stimulating immune response and direct tumor ablation, could be mentioned in the paper( https://doi.org/10.1016/j.apsusc.2021.150795 , https://doi.org/10.1016/j.jphotobiol.2019.111587 ).

Author Response

1) The review “Patient-derived lung tumoroids an emerging technology in drug development and precision medicine” explores the state of the art and the applications of lung tumoroids in clinical research. The organization of the review is very schematic, allowing a comprehensive understanding of the subject. I do recommend this paper for publication after minor revision. Resolution of figures 1 and 4 could be improved;

Answers : Figure 1 has been enlarged, and its resolution was improved, enabling scaling up to 500% without pixelisation. Figure 4 is an original figure, produced for this article via the Tayasui sketch graphic software. The resolution was optimal before pasting into Word file.

2) A table including the state of the art in integrative microfluidics could help focusing on current research;

Answers : A new table was created (Table 4), summarizing the progress of microfluidic systems in lung cancer, with 5 initial references + 1 reference that was not cited in the previous version.

3) The section in which immunotherapy is discussed (7) should be expanded;

Answers : This section was updated with more details on the different immunotherapies treatments approved in NSCLC, the mechanism of action, intrinsic resistance mechanism, and the contribution of Immunoscore. The proposed modifications appears in blue in the uploaded version.

4) The use of photothermal therapy, which is a non-invasive technique capable  of stimulating immune response and direct tumor ablation, could be mentioned in the paper( https://doi.org/10.1016/j.apsusc.2021.150795 , https://doi.org/10.1016/j.jphotobiol.2019.111587 ).
Answers : We thank you for this interesting input, we added this reference in section 6.

Reviewer 2 Report

Manuscript ID:  biomedicines-1777604  -  Review

 Title: Patient-derived lung tumoroids an emerging technology in drug development and precision medicine

 In this very interesting review the authors give a quite wide overview on tumoroid methodology, in particular focusing on lung carcinoma.

 The review is very well written and organized and allows the reader to have a very up-to-date panorama on this field but, at the same moment, point out clearly what are at present the limits of this kind of research.   

 Minor point.

In my opinion, it may be useful to better clarify the definition of organoid/tumoroid/spheroid when introducing table 2, adding a couple of sentences in par. 4. The nomenclature is actually a bit confusing and may be of great help to better define the real differences between the different models.

Author Response

In this very interesting review the authors give a quite wide overview on tumoroid methodology, in particular focusing on lung carcinoma.
The review is very well written and organized and allows the reader to have a very up-to-date panorama on this field but, at the same moment, point out clearly what are at present the limits of this kind of research.
Minor point.
In my opinion, it may be useful to better clarify the definition of organoid/tumoroid/spheroid when introducing table 2, adding a couple of sentences in par. 4. The nomenclature is actually a bit confusing and may be of great help to better define the real differences between the different models.

Answers : We agree that nomenclature is important, and certainly deserves clarification as well as harmonization. We did not feel allowed to change the names given by authors to their technologies in Table 2, but a novel paragraph was added after this table to provide our set of definitions for spheroids, organoids, or tumoroids.